# CanvasEmb: Learning Layout Representation with Large-scale Pre-training for Graphic Design

## Abstract

Layout representation, which models visual elements in a canvas and their inter-relations, plays a crucial role in graphic design intelligence. With a large variety of layout designs and the unique characteristic of layouts that visual elements are defined as a list of categorical (e.g. shape type) and numerical (e.g. position and size) properties, it is challenging to learn a general and compact representation with limited data. Inspired by the recent success of self-supervised pre-training techniques in various natural language processing tasks, in this paper, we propose CanvasEmb (Canvas Embedding), which pre-trains deep representation from un-labeled graphic designs by jointly conditioning on all the context elements in the same canvas, with a multi-dimensional feature encoder and a multi-task learning objective. The pre-trained CanvasEmb model can be fine-tuned with just one additional output layer and with a small size of training data to create models for a wide range of downstream tasks. We verify our approach with presentation slides data. We construct a large-scale dataset with more than one million slides, and propose two novel layout understanding tasks with human labeling sets, namely element role labeling and image captioning. Evaluation results on these two tasks show that our model with fine-tuning achieves state-of-the-art performances. Furthermore, we conduct a deep analysis aiming to understand the modeling mechanism of CanvasEmb, and demonstrate its great potential use on more applications such as layout auto completion and layout retrieval.

## 1 Introduction

Graphic design leverages layout to set up and arrange visual elements in a canvas for conveying message in different types of documents, while layout representation is the reversed process to understand visual elements and their inter-relations in a canvas, which is the key for the analysis (Stoffel et al., 2010), retrieval (Beusekom et al., 2006) and generation (Li et al., 2020b; Lee et al., 2020) of graphic designs. However, elements in a layout are complex, which are defined with multi-dimensional properties such as type (e.g., text box, image or button), position and color. For example, the web page and presentation slide shown in Figure 1 is defined by a lot of settings, as each example is constructed by several elements and each element is defined by several proprieties. Due to the complex and sparse features of elements, as well as the rich diversity of layouts, learning a general and compact layout representation is challenging with limited amount of data.

Previous works related to layout representations (Li et al., 2019; Tabata et al., 2019; Lee et al., 2020) are mostly task-oriented. They simplify the layout only as the positions of elements, and directly optimize task-specific labels with less than a few thousands instances. Recently a majority of self-supervised pre-trained models such as ELMO (Peters et al., 2018), GPT (Radford, 2018) and BERT (Devlin et al., 2019) have shown promising results in improving a variety of natural language processing (NLP) tasks. The success of pre-trained models in NLP has inspired us to learn contextual layout representations from large-scale unlabeled graphic designs, which can facilitate various downstream tasks for design intelligence. As one highly related work, LayoutLM (Xu et al., 2019) is a document pre-trained model incorporating both text content and layout information for scanned documents. However, it is difficult to generalize to other document types, since its input is

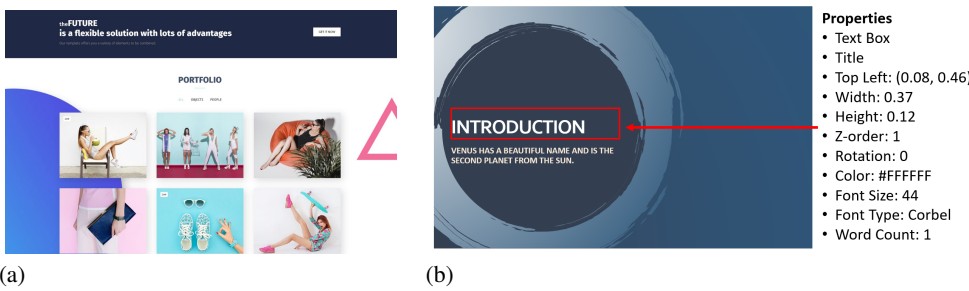

(a)               (b)

Figure 1: Example layouts for different document types: (a) Web Page (b) Slide. As shown on the right, elements have multiple properties, which make layout complex and diverse.

word-level and it defines layout only as the word position, which is insufficient to describe a layout in graphic design.

In this paper, we present CanvasEmb, a large-scale pre-trained model for learning contextual layout representation. It is designed to pre-train deep representation from unlabeled graphic designs by jointly conditioning on all the context elements in the same canvas, and the pre-trained CanvasEmb model can be fine-tuned with just one additional output layer and with a small size of training data to create models for a wide range of downstream tasks. Specifically, we define a generic and high-coverage vocabulary to describe element properties in the canvas. A feature encoder is designed to jointly incorporate multi-dimensional properties, and it is developed with the multi-layer Transformer (Devlin et al., 2019) for modeling element contexts. To ensure the representation conditioning on all dimensions of element contexts, we adopt the masked language modeling strategy with a multi-task objective, where we randomly mask some properties of elements for prediction in the pre-training.

To verify our approach, we construct a large-scale dataset with more than one million presentation slides containing rich layout meta-information for pre-training. We then propose two novel downstream tasks for layout understanding with human labeling sets to evaluate the performance of our pre-trained CanvasEmb model. The first task is element role labeling. Only given the information of layout, the goal is to classify the semantic role of each element (e.g., title, subtitle). The second task is image captioning, which detects if a text box and an image in a slide belongs to the image captioning relation. Experimental results on the two tasks show that fine-tuning the CanvasEmb model achieves state-of-the-art performance. Furthermore, we conduct deep analysis to understand the modeling mechanismCanvasEmb. Also, we demonstrate the great potential use of our pre-trained CanvasEmb with two extended applications, including layout auto completion (Li et al., 2020b) and layout retrieval.

The contributions of this work are as follows:

- We propose CanvasEmb, which to the best of our knowledge is the first pre-trained model for layouts in graphic design. It can be fine-tuned with a small size of training data for a wide range of downstream tasks.

- We construct a large-scale dataset of presentation slides with rich layout information, as well as two novel tasks for layout understanding (i.e., element role labeling and image captioning) with human labeling sets.

- We demonstrate that our model achieves state-of-the art performances on the two downstream tasks, and show the potential for more applications such as layout auto-completion and layout retrieval.

## 2    RELATED WORK

Layout representation is the focal point of design in rich media, including presentation slides, magazines, comics, posters and web pages. High-quality representations can be conductive to multiple practical design tasks. Early works on design layout or document layout mainly rely on templates (Hurst et al., 2009; Damera-Venkata et al., 2011) or heuristic rules (O'Donovan et al., 2014; Tabata et al., 2019) and require professional knowledge and manual efforts. To efficiently facili-

tate the problem-solving aspects of sketching in the graphic designs, Todi et al. (2016) propose an interactive layout design tool which uses a real-time layout optimiser without requiring extensive input. However, these methods are restricted and usually fail to model the rich varieties of media information. Xinru Zheng & Lau (2019) make use of the content information to model the graphic design layouts in the purely data-driven scenario to adapt to the contents to be laid out.

Recently, there is a trend to adopt neural networks and deep learning methods to promote automating layout to be more efficiently. For example, to be more user-friendly, Pang et al. (2016) adopt attention mechanisms to trace the user's attention, and Lee et al. (2020) improve the conventional GAN-based methods (Li et al., 2019; Xinru Zheng & Lau, 2019) to explicitly model relationships among components and user-specified constraints. BERT4Rec (Sun et al., 2019) employs the deep bidirectional self-attention to model user behavior sequences. And Li et al. (2020b) develop Transformer-based tree decoders on the task of auto completion of user interface layout design, which can ease the efforts of UI designers and developers.

However, previous works typically deal with limited kinds of design elements and fail to give general and scalable solutions to layout representation learning. Enlightened by the significant impact of large-scale pre-trained models in the area of NLP (Peters et al., 2018; Radford, 2018; Devlin et al., 2019; Yang et al., 2019) and multi-modal learning (Sun et al., 2019; Lu et al., 2019; Li et al., 2020a), our work implements the attention-based Transformer framework enhanced with pre-training to propose a data-driven and scalable method that captures contextual information for layout representation, which can be applied well on downstream tasks in graphic design.

## 3 LAYOUT IN GRAPHIC DESIGN

Layout in graphic design refers to the way in which we arrange the visual elements on a canvas. Though some settings might vary specific to different document types, there exists basic characteristics of elements that make up the content of layouts (example shown in Figure 1):

- **Type Properties**. Elements can be text boxes, pictures or lines. According to the semantic roles, elements can be divided into title, subtitle, button or other placeholders.
- **Geometry Properties**. Position and size indicate the elements' placement in the layout. Besides, z-order is the ordering of overlapping two-dimensional elements, and rotation describes the an element's circular movement.
- **Color Properties**. Color is one of the most straightforward visual feature, including the RGBA channels and extra features such as color gradient.
- **Content-related Properties**. Though user contents are separated from the layout, some content-related properties (e.g., text font size and font type) can affect the layout arrangement.

Elements are complex and sparse, composed with the above properties of either categorical (e.g., shape type, color) or numerical (e.g., position, font size, word count) values. Hence, layouts are diverse and complicated for modeling. In the next section, we will introduce our approach for layout representation learning.

## 4 MODELING

We present our model CanvasEmb, which inputs elements with multi-dimensional properties and outputs representation for the layout. To train our model, we adopt the two-stage learning framework, namely pre-training and fine-tuning.

### 4.1 MODEL ARCHITECTURE

We formulate the input as a sequence of visual elements $\{\mathbf{x}_0, \mathbf{x}_1, \mathbf{x}_2, ..., \mathbf{x}_n\}$ in the layout, where each element $\mathbf{x}_i$ is defined with $m$ properties $\{\mathbf{p}_i^1, \mathbf{p}_i^2, ..., \mathbf{p}_i^m\}$. Here $\mathbf{x}_0$ is the sequence representation which is randomly initialized. Figure 2 shows the overview architecture of our model, which is similar to BERT (Devlin et al., 2019). The feature embedding encodes high dimensional properties for elements, and is concatenated with the transformer encoder to model the global context

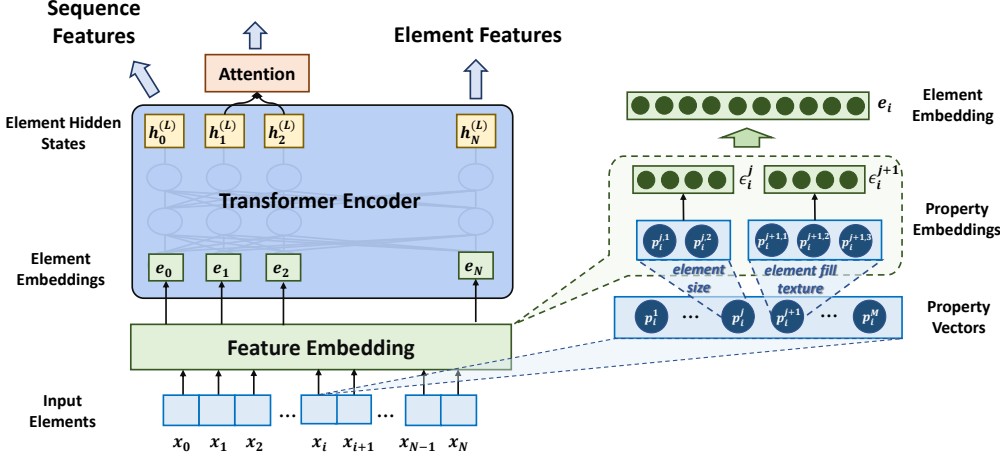

Figure 2: Overall Architecture of CanvasEmb.

of elements. The output representation can be further used to make prediction of element-level and layout-level labels, as well as the relations between the elements, with an extra task-specific prediction layer. Here, we introduce the details of model components.

**Feature Embedding.** For the $i$-th element $\mathbf{x}_i = [\mathbf{p}_i^1; \mathbf{p}_i^2; ...; \mathbf{p}_i^m]$, the embedding $\mathbf{e}_i$ is obtained by concatenating $m$ property embeddings:

$$\mathbf{e}_i = \Theta(\epsilon_i^1 \oplus \epsilon_i^2 \oplus ... \oplus \epsilon_i^m), \tag{1}$$

where $\oplus$ is the concatenation operator and $\Theta$ is a non-linear transform function.

For each channel $j$, the corresponding property $\mathbf{p}_i^j$ contains multi-dimensional values. For example, given a 2-*dim* numerical property $\mathbf{p}_i^j = [p_i^{j,1}; p_i^{j,2}]$ (e.g., element size with height and width), the embedding in this channel can be calculated as:

$$\epsilon_i^j = \xi^j(p_i^{j,1}) \oplus \xi^j(p_i^{j,2}), \tag{2}$$

where $p_i^{j,k}$ represents the $k$-th value in $\mathbf{p}_i^j$ and $\xi^j$ is the embedding function.

There are two types of embedding functions. For properties with categorical values such as type and color, we use the embedding matrix as the learning parameter. For properties with numerical values such as position and size, the positional encoding (Vaswani et al., 2017) is adopted:

$$PE_{(p_i^{j,k}, 2h)} = \sin(p_i^{j,k}/10000^{2h/d^{j,k}}) \tag{3}$$

$$PE_{(p_i^{j,k}, 2h+1)} = \cos(p_i^{j,k}/10000^{2h/d^{j,k}}) \tag{4}$$

where $d^{j,k}$ means the embedding dimension assigned to $p_i^{j,k}$.

**Transformer Encoder.** On top of the feature embeddings, we use a transformer encoder (Vaswani et al., 2017) to encode the element contexts. Similar to BERT (Devlin et al., 2019), the multi-layer transformer with the multi-head self-attention mechanism enables to capture correlations between different elements and property fields. Finally, we can get the low-dimensional representations $\{\mathbf{h}_0^{(L)}; \mathbf{h}_1^{(L)}; ...; \mathbf{h}_n^{(L)}\}$ for all elements from the last, *i.e.* the $L$-th encoding layer.

### 4.2 PRE-TRAINING CANVASEMB

We adopt the masked language modeling (MLM) as the objective of CanvasEmb pre-training, which has been proven effective in several domains (Devlin et al., 2019; Sun et al., 2019; Lu et al., 2019). To enable the model to learn correlations from different properties, we randomly select any one type of properties to mask for an element during training. CanvasEmb is trained to approximately maximize the pseudo likelihood $E_{\mathbf{x} \sim \mathcal{D}} \sum_{i=1}^n \sum_{j=1}^m P(\mathbf{p}_i^j | \mathbf{p}_i^{\backslash j}, \mathbf{x}_{\backslash i}; \theta)$ (Sun et al., 2019), where $\mathcal{D}$ is the real data distribution, $\theta$ represents the model parameters, $\mathbf{p}_i^{\backslash j}$ represents the properties masking $p_i^j$ of

the $i$-th element, and $\mathbf{x}_{\setminus i}$ means all the elements in the input excluding for the $i$-th one. Particularly, we adopt *cross-entropy* loss and *MSE* loss for classification and regression tasks respectively:

$$\mathcal{L}(\theta) = \begin{cases} \sum_{i=1}^{n} \sum_{j \in \mathcal{M}_c} -\log P(\mathbf{p}_i^j | \mathbf{p}_i^{\setminus j}, \mathbf{x}_{\setminus i}; \theta) \\ \sum_{i=1}^{n} \sum_{j \in \mathcal{M}_u} ||\mathbf{p}_i^j - \hat{\mathbf{p}}_i^j||_2 \end{cases} \tag{5}$$

Here $\mathcal{M}_c$ and $\mathcal{M}_u$ represent the index sets of categorical and numerical properties, $\hat{\mathbf{p}}_i^j$ is the model prediction.

### 4.3 FINE-TUNING CANVASEMB

For different downstream tasks, we feed the same input as in the pre-training phase, and fine-tune the model with extra task-specific layers:

**Element-Level** tasks are aimed to predict specific features of the elements, *e.g.* properties which are different from but correlated to the input ones.

**Element-to-Element** tasks predict the relations between a pair of elements. In CanvasEmb, we build a query-memory attention-based mechanism for the relation prediction (Wang et al., 2017):

$$\mathcal{R}(\mathbf{x}_i, \mathbf{x}_j) = \alpha_{i,j} + \alpha_{j,i} \tag{6}$$

$$\alpha_{i,j} = \frac{\exp W_Q^r \mathbf{h}_i^{(L)} \cdot W_M^r \mathbf{h}_j^{(L)}}{\sum_{j'=0}^{n} \exp W_Q^r \mathbf{h}_i^{(L)} \cdot W_M^r \mathbf{h}_{j'}^{(L)}} \tag{7}$$

where $W_Q^r$ and $W_M^r$ represent trainable transforming weight matrix.

**Sequence-Level** tasks utilize the $\mathbf{x}_0$ representation token to make prediction.

## 5 EXPERIMENT

### 5.1 DATASET AND TASK DEFINITION

**Pre-training Dataset.** CanvasEmb is designed to represent various types of graphic designs. In this paper, we adopt presentation slides as an example, as a huge amount of slides are publicly available and the understanding of slides would enable many applications. We first crawl presentation slides from the web to conduct a pre-training dataset with one million pages of slides[1]. For each element (e.g., a textbox or a table) in a page of the slide, we further extract their properties as defined in Section 3.

**Element Role Labeling.** This task is to detect the semantic role of each element on a canvas, which is one of the most important problems in Robotic Process Automation. As shown in Figure 3(a), each element is assigned with one of the following labels: *title*, *subtitle*, *footer*, *decorator*, *heading*, *caption*, and *other*. We create a dataset containing $48,309$ slides and split it into $38,819$ for training and $9,490$ for testing. Each element on the slides is annotated by five labelers and we use majority voting to assign the final label.

**Image Captioning.** Given a pair of an image and a text box in a slide, the goal is to detect if the pair belongs to image captioning or not. In the example shown in Figure 3(b), there are four caption pairs on the right of the slide. With similar annotation pipeline, we finally get $2,996$ slides, divided as $1,496$ for training and $1,500$ for testing.

### 5.2 SET UP

**Model and Training Parameters**: Following Devlin et al. (2019), we testify the effect of model size by comparing two model sizes: CanvasEmb (Base) (L=3, H=64) and CanvasEmb (Large) (L=6,

---

[1] We plan to open source the script for crawling slides.

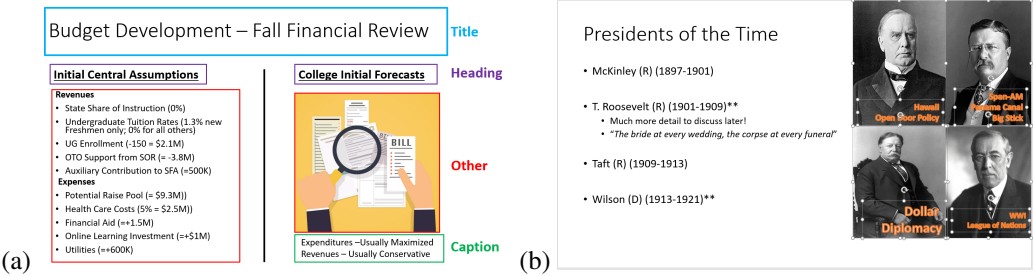

Figure 3: Examples for two layout understanding tasks. For (a) element role labeling, element in blue color indicates *title*, purple for *heading*, green for *caption* and red for *other*. For (b) image captioning, four images each with text box in orange color are image caption pairs.

H=128), where $L$ and $H$ denote the number of encoder layers and the hidden size, respectively. And the sequence length during training is 16 for both models. During pre-training, we randomly select one type of property to mask for each sample by replacing it with [*MASK*] (90% of the time) or random values (10% of the time). For multi-task learning objectives, the loss of each prediction task is weighted according to the current performance on the validation set, where the weight for task $p \in \mathcal{M}_c$ is $\log \max(\frac{1}{accuracy_p + \epsilon}, 1)$ and for task $p \in \mathcal{M}_u$ is $\log \min(\frac{squared\_error_p}{10}, 100)$. For fine-tuning, we adopt the $\alpha$-balanced focal loss (Lin et al., 2017) to address the label unbalance. The trade-off parameters $\lambda$ in focal loss is 2. All models are trained using Adam (Kingma & Ba, 2015) with mini-batch size 64. The learning rate is initially set to $10^{-5}$ and adaptive learning rate decay applied. We also adopt early stopping and use gradient clipping (Pascanu et al., 2012).

**Baselines**: Since there are few strong neural methods that are applicable for the graphic layout design, we compare our proposed model CanvasEmb against the traditional method, *i.e. Decision Tree* implemented as GBDT (Ke et al., 2017). The input is the same as CanvasEmb and we use the same focal loss settings.

## 5.3 RESULT ANALYSIS

Table 1: Results for element role labeling.

| 100% labeled data | title | subtitle | footer | heading | decorator | caption | other | **macro** | **micro** |
|---|---|---|---|---|---|---|---|---|---|
| Decision Tree | 96.62 | 82.28 | 93.24 | 53.79 | 58.45 | 50.34 | 73.59 | 76.79 | 89.34 |
| CanvasEmb (Base) | 95.65 | **91.76** | **95.94** | 34.55 | 96.92 | 71.15 | 71.48 | 82.41 | 91.77 |
| - w/o pretrained | 95.88 | 90.62 | 95.76 | 39.79 | 97.44 | 64.91 | 72.73 | 81.95 | 91.99 |
| CanvasEmb (Large) | **98.74** | 88.28 | 94.18 | **55.15** | **98.66** | **78.25** | **78.73** | **85.45** | **95.97** |
| - w/o pretrained | 95.18 | 88.07 | 92.37 | 30.93 | 97.29 | 70.06 | 65.59 | 79.93 | 90.87 |
| **30% labeled data** | | | | | | | | | |
| Decision Tree | 97.06 | 71.44 | 66.67 | 0.00 | 63.03 | 19.75 | **36.14** | 54.54 | 86.62 |
| CanvasEmb (Base) | 97.69 | **89.78** | 88.20 | 60.56 | 95.40 | 75.80 | 29.18 | 80.51 | 93.16 |
| - w/o pretrained | 93.67 | 86.61 | 82.83 | 48.44 | 95.71 | 58.03 | 19.16 | 74.77 | 86.88 |
| CanvasEmb (Large) | **98.98** | 87.46 | 88.20 | **61.59** | 95.40 | **75.80** | 29.18 | **82.52** | **95.42** |
| - w/o pretrained | 94.08 | 88.18 | **92.51** | 51.73 | **95.89** | 73.08 | 15.89 | 78.55 | 88.18 |

**Element Role Labeling.** We report the F1 score for each role and the overall macro/micro F1 metrics in Table 1. On the one hand, all CanvasEmb methods outperform traditional Decision Tree model with a large margin on most of the metrics. On the other hand, CanvasEmb (Large) achieves better results than CanvasEmb (Base), and pre-training further boosts the performances.

**Image Captioning.** We report F1 and AUC scores in Table 2. Similarly, CanvasEmb (Large) with pre-trained achieves the best results. Since this task focuses on the relation between elements, we also train an enhanced baseline (+2D features) where we explicitly incorporate additional hand-crafted 2-dimensional features, such as the distance between two elements. We observe that CanvasEmb (Large) outperforms the enhanced baseline with a large margin. This indicates the ability of CanvasEmb to capture the relations between elements, which is much more efficient than the heuristic way of manual feature engineering.

| 100% labeled data | F1 | AUC |
|---|---|---|
| Decision Tree | 84.30 | 98.32 |
|   + 2D features | 84.50 | 98.21 |
| CanvasEmb (Base) | 88.99 | 98.23 |
|   - w/o pretrained | 89.28 | 98.33 |
| CanvasEmb (Large) | **90.95** | **98.51** |
|   - w/o pretrained | 89.68 | 98.32 |
| **30% labeled data** | | |
| Decision Tree | 76.39 | 96.42 |
|   + 2D features | **81.34** | **97.36** |
| CanvasEmb (Base) | 78.37 | 94.26 |
|   - w/o pretrained | 75.69 | 93.69 |
| CanvasEmb (Large) | 80.64 | 95.27 |
|   - w/o pretrained | 79.43 | 94.85 |

Table 2: Results for image captioning.

Table 3: Results for auto completion. We show accuracy for Shape Type (ST.) and Fill Color (FC.), IoU for Position (P.).

| Model | ST. | FC. | P. |
|---|---|---|---|
| CanvasEmb (Base) | 87.38 | 69.49 | 44.01 |
| CanvasEmb (Large) | 89.37 | 75.63 | 46.40 |

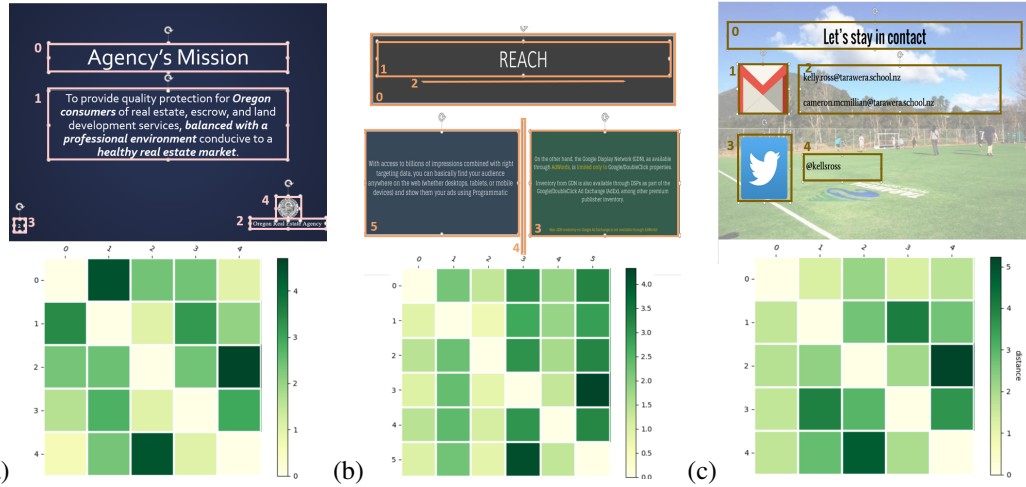

(a)          (b)          (c)

Figure 4: The element-to-element relations. The first row shows the original slides with element index, and the second row shows the corresponding headmaps which X- and Y-axis are the element indexes. (a) Pre-Trained Model (b) Element-Role-Labeling Model (c) Image-Captioning Model.

## 5.4 EFFECT OF PRE-TRAINING

**Correlation between pre-training and model size.** In general, CanvasEmb models with pre-trained outperform those without pre-trained on both tasks. And the increase of model size in the large model (L=6, H=128) further boosts the performance compared to the base model (L=3, H=64). For example, in the task of element role labeling, the pre-trained model brings 5.52% macro F1 gain for CanvasEmb (Large) while only 0.46% gain for CanvasEmb (Base). This is accordant with our intuition, *i.e.* larger model can capture and embody more knowledge from the pre-training.

**Correlation between pre-training and labeled fine-tuning data size.** We also conduct ablation study with only 30% labeled training data for both downstream tasks, as shown in Table 1 and 2 respectively. We observe that pre-training contributes more when labeled data size is small. In element role labeling task, there is $\triangle = 5.47\%$ macro F1 gain from pre-training for CanvasEmb (Base) under the setting of 30% training data, compared to only 0.46% gain using full data. Similarly, in the task of image captioning, there is $\triangle = 2.68\%$ F1 increase for CanvasEmb (Base) from pre-training using 30% data while pre-training only works a little bit for the base model when using full data. This shows that our pre-trained models are robust and well aligned to the downstream tasks, which alleviates the data scarcity issue.

## 5.5 PROBING INTO SHAPE CONTEXT

To investigate how CanvasEmb captures element-to-element relations, we use the impact function (Wu et al., 2020) $f_{impact}(\mathbf{x}_i, \mathbf{x}_j) = dist(\mathbf{h}_i(\mathbf{X}\backslash\{\mathbf{x}_i\}), \mathbf{h}_i(\mathbf{X}\backslash\{\mathbf{x}_i, \mathbf{x}_j\}))$ to show the importance

Figure 5: (a) A query layout and (b)(c) top 2 similar candidates retrieved by CanvasEmb (Large).

of the elements by calculating the distance between the hidden states of the layout when removing elements $\{\mathbf{x}_i\}$ and $\{\mathbf{x}_i, \mathbf{x}_j\}$. The impact matrix heatmaps are displayed in Figure 4.

• **Pre-Training**. From Figure 4(a), we observe that elements in close region tend to have stronger influences on each other (shown by darker-colored pixels). For example, element 2 and 4 on the bottom right of the slide have significantly strong impacts on each other, but weaker impacts on element 3 on the left. This indicates that the pre-trained model can typically capture location information in layout representation.

• **Element Role Labeling**. Figure 4(b) visualizes the model fine-tuned on the task of element role labeling. We can see elements 1, 3, 5 generally have the most significant impacts on all other elements in the slide. Specifically, element 3 and 5, which are symmetric and have the same semantic roles, strongly affect each other. This shows that the model learns representations mainly based on the content-related features and location of elements in the slide.

• **Image Captioning**. For the image captioning task in Figure 4(c), the fine-tuned model pays more attention to images and the corresponding caption-like elements. Element pairs $\{1, 3\}$ and $\{2, 4\}$ have very strong impacts on each other, while element 0 has the weakest influences in the slide. This is an interesting observation to see how caption pairs in a slide interact with each other, as the model in this case obviously separates the element 0 from others with captioning relation.

### 5.6 EXTENDED APPLICATIONS OF PRE-TRAINED CANVASEMB

To further demonstrate the knowledge embodied in our pre-trained CanvasEmb, we show two extended applications.

**Layout Auto Completion.** Given a partial layout, the model needs to complete the layout by predicting the remaining elements with their properties (Li et al., 2020b). We constrain the setting to one remaining element, which is aligned to our pre-training objective. From the result shown in Table 3, we can see our pre-trained model without any fine-tuning achieves promising results with respect to the properties of shape type, fill color and position. This shows the accurate modeling of elements and contextual representations of layouts in the pre-trained CanvasEmb.

**Layout Retrieval.** Given a query layout, the scenario is to retrieve similar layouts from the database, which can be used for layout recommendation. Figure 5 shows an example retrieved by CanvasEmb (Large) with layout embedding cosine similarity. As we can see, both the query and the two candidates contain similarly four images with captions. For a quick evaluation, we construct 20 queries and manually score model's retrieval quality (rate similarity score from 1 to 5). The average score of top 3 candidates are 3.86 ($> 3$ above average), which demonstrates CanvasEmb effective layout representations.

## 6 CONCLUSION

In this paper, we present CanvasEmb, a large-scaled pre-trained model for layout representation in graphic design. It encodes elements in a layout with the multi-dimensional properties and models the element context with a Transformer-based architecture. To pre-train our model CanvasEmb, we adopt the masked language modeling with multi-task learning objective. In the two proposed tasks related to layout understanding, fine-tuned CanvasEmb achieves state-of-the-art performances. Furthermore, we conduct a deep analysis to prob into the embodied knowledge learned by CanvasEmb, and show its great potential for more applications in graphic design. As future work, we are going to apply our model to more downstream tasks, and verify our model on more document types such as web pages or word documents.

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
