# OpenReview forum: "CANVASEMB: Learning Layout Representation with Large-scale Pre-training for Graphic Design"
_ICLR.cc/2021/Conference — Reject_

### Official Review · AnonReviewer3 · 2020-10-28
**BERT applied to graphic design layouts (nice!). Requires more experimenting and more details need to be presented**

**Rating:** 4
**Confidence:** 3

**Review:**

This paper proposes to learn representations for graphic design layouts unsupervised using a Masked Language Model objective as in BERT with additional domain specific input embeddings. It also presents a large dataset of slides along with manual annotations used for their evaluation.

The idea of using BERT like training to pre-train features for graphic design layouts is sound and the proposed approach modifies the input token representation to represent different layout elements -- using positional embeddings for continuous attributes (position, rotation etc.) and learned embeddings for categorical attributes (element type etc.). Since there is no positional embedding for position in the sequence, the ordering in the sequence does not matter for training, which is a huge benefit over training sequential models in this context, where ordering can be arbitrary.

I believe the evaluation of the method presented in the paper is lacking and could be improved significantly. Moreover, it would be useful to make it clear if the authors plan to release the collected slides dataset since that would count as a positive contribution in my books. More training details (how many/what GPUs, how much time/epochs, any learning rate tricks, optimizer etc.) would also help with the reproducibility aspect of the paper.

W.r.t evaluation, the authors propose to do the tasks of element type classification (called role labeling in the paper) and image-caption pair detection (i.e. a binary classifier). On both of these tasks, the only baseline presented is a gradient boosted decision tree, which performs quite well on both tasks already. This begs the question, are all these parameters used to learn CanvasEmb needed? In my head there are the following ways to answer this question:
1. Are there other higher complexity tasks (where the decision tree baseline would fail a lot) that are important for graphic design layout intelligence?
2. Are there other lower complexity (parameter wise) methods that also take 2D layout into consideration that you could train for these two tasks? For example, how would a simple convnet trained to classify UI elements perform? You could imagine a variety of ways of showing the convnet which element to classify, one example would be passing its bounding box mask as a separate input channel. How does CanvasEmb perform as compared to them?

The other evaluation presented is on layout retrieval and layout auto completion. Layout auto completion in this case is a misnomer (when related to previous work), since the authors only mask 1 attribute of 1 element at a time and compute the accuracy of predicting that attribute on test set examples. This differs strongly from layout auto completion in other work where complete elements along with their properties are added. Similarly for layout retrieval, a quick ad-hoc rating study is presented.

It is hard to position this work w.r.t. previous work given how the evaluation is performed. It would be great to evaluate this method in the context of existing work to really understand the importance of doing this sort of pre-training. Doing higher complexity tasks as mentioned above would also be very useful. For example, can you use CanvasEmb to offer layout suggestions (as in DesignScape, CHI 2015 as an example)? You can imagine a Gibbs Sampling like procedure where you repeatedly randomly erase a part of the parameters of the layout elements (i.e. their position, rotation, color etc.) and repredict them for some steps. Would this be able to generate plausible layout suggestions?

Overall, I believe this is an exciting direction and I'm looking forward to hear how the authors think their paper could be evaluated on harder tasks, against stronger baselines and against existing work.

---

> ### Author Response · Authors · 2020-11-23
> **Thanks for your comments.**
>
> As most reviewers mentioned, the key issues of this paper are the evaluation with previous works and more formal evaluation methods. We will improve these in the next version.
>
> -Evaluation of higher complexity tasks: The current two tasks are motivated by real downstream applications for layout recommendations (one for element classification and one for element relation classification). We might think of more difficult tasks related to layout understanding in the future.
>
>
> -Compare with other lower complexity methods / existing works: yes, we will add extra baselines of existing works for comparison. Some of them contain simple neural architecture, which will further demonstrate if such Transformer and pre-training works better.
>
>
> -Evaluation of layout auto completion and layout retrieval: The two applications are used as extra evaluations for the pre-trained model. We simplify the setting of layout auto completion task as to align with our pre-training tasks. According to your suggestion, we will try to follow the same setting of previous works (predict only the geometry properties and not constrained to only 1 element for prediction). For the layout retrieval evaluation, since there is no existing dataset for evaluation, we just conduct a simple human evaluation. We will design more systematic evaluation in the next version.

---

### Official Review · AnonReviewer2 · 2020-10-28
**Different Approach on the non-trivial task of Layout representation, but unsatisfactory evaluation**

**Rating:** 5
**Confidence:** 3

**Review:**

Summary of the work:
This work presents a Transformer-based framework to learn Graphical Layout representation/embedding, taking inspiration from transformer-based models in the area of NLP.  To train their model, named CANVASEMB, the paper contributes a dataset of Powerpoint Slides, with more than 1 million slides (the paper promises to make the dataset public). When performing new tasks on graphical layouts, CANVASEMB can be used as a pre-trained model. The paper also demonstrates that CANVASEMB achieves SOTA performance on two downstream tasks, viz., element role labeling, and the so-called "image captioning" in the context of a layout image and its text tag.

Originality:
The task being attempted (Learning Layout embeddings) is not new. However, the paper makes use of a Transformer-based model for Layout embedding, which is different and was not explored/employed before. However, the idea of employing Transformers is not well motivated;  the method section is technically sparse, in the sense that not much of the Transformer model is presented (at least using a Figure, if not via Mathematical formulation). It would help the reader if details on the Transformer model and the Attention scheme were presented to some extent.

Significance:
Layout Representation and Learning is an import problem and is gaining significance. That said, any method should evaluate thoroughly to demonstrate its merits and advantages over prior works, for potential impact. This paper lacks a thorough evaluation. More on this later.

References:
Penultimate line in the first paragraph of RW section: It should be "Cao et. al 2019", not "Xinru Zheng and Lau (2019)". The first author naming convention cannot be superseded at the writer's discretion. Only include the second and last author in the reference? Has been cited again after this, in the same way. I will assume that this was an oversight. Needs to be corrected.
Also, the paper is missing the following reference, which, similar to Li 2020b, uses a Tree-based representation of Document Layouts.
1) READ: Recursive Autoencoders for Document Layout Generation, CVPR 2020


Quality:
1) The paper claims that CanvasEmb is the first work to provide pre-trained models for layouts in graphic design. I think the "Content-Aware Generative Modeling of Graphic Design Layouts" from SIGGRAPH 2019 is the first work to do so, although on a much smaller scale (~4K magazine designs as against 1M used in this paper).
2) The paper does not contrast how it is different from the above work (Cao et al., SIGGRAPH 2019). What are the key differences? What is special about CANVASEMB, that can not be done by the 2019 work when given such a large dataset? Is there a key technical bottleneck that is overcome in this work? I would first like to know the maximum number of layout elements in a Slide in the entire dataset.
3) The "Related Work" section could be better. Currently, there seems to be a lack of some sort of explanation on what differentiates this work from the prior works. In other words, there is a frequent and incoherent jump from reference-to-reference in the context of this paper. The desire for a reader is to understand the difference in contributions of this work as compared to the prior works in the related work section.
4) "Content-Related Properties" are already taken into account when the semantic box geometries are considered. So, the text font-size and font-type will not have an additional effect on the layout, but only on the perceptual quality of a content-filled layout. So, I don't think this counts as one of the layout properties when you have the box geometries already accounted for.
5) The so-called "image captioning" task is nothing but a binary classification of an image-text tag pair.
6) Comparisons against the SIGGRAPH 2019 paper (Cao et al, 2019), Li 2020b, and READ (CVPR 2020) should be presented.  Doing such an evaluation on the same datasets as CANVASEMB will demonstrate the strength and weaknesses of the Transformer-based architecture.
7) In addition to these evaluations, I would like to see why the SOTA Graph Neural Networks can not be employed on this task when they can capture rich structural properties of a layout and require less computation compared to Transformers. This is a very important and interesting experiment, which should be performed.  Moreover, training a Graph Neural Network does not require humungous data. This weakens the motivation of the approach presented in this paper.

Clarity:
1) The writing is good. As I said earlier, the RW section could (and should) be improved.
2) I would suggest presenting an image that contains the components of the Transformer. The reader should not have to refer to the cited papers to know/learn what the Transformer is made up of.
3) I think the word "pre-trains" in the paper means to say that "CanvasEmb" can be used as a pre-trained model for other tasks. CanvasEmb, in the due course of its development, is not "pre-trained" to begin with. So, I think the sentence in the abstract, the introduction, and parts of the evaluation involving the word "pre-trains" should be rephrased to remove confusion and for an easier understanding of the paper. Ex: A Transformer is first trained using the Slides crawled from the internet, which we term as CanvasEmb, and can be used as a pre-trained model for other downstream tasks.
4) "mechanism" typo in the last paragraph of introduction (it appears twice continuously)
5) In Equation 5, the tasks for each loss formulation should be written (in the equation itself)
6) I would like to know the maximum number of layout elements in a Slide in the entire dataset. For ex: if there are 3 Text Boxes and 2 Figures in a Slide, then the total number of layout elements is 5.
There has been no mention of this in the entire paper, but this is crucial to understand if there is really a need for Transformer-based architecture.

Pros:
1) Different, interesting approach to Layout Embedding
2) Contributes a huge dataset of Powerpoint Slides (1M)
3) Simple and easy writing


Cons:
1) Some important dataset-related stats are missing (as pointed above). They are essential for understanding the richness of the dataset and to make an informed decision on what kind of deep-learning models should be employed when using such a dataset for different tasks.
2) Weak evaluation
3) Comparison to Cao et al., SIGGRAPH 2019, Li et al 2020b, and READ (CVPRW 2020) are missing. The last two are tree-based generative models of Document layouts, and should be tested for performance on the same dataset used to train CANVASEMB.
4) Comparison to SOTA Graph Neural Networks: Missing explanation for a motivation to use Transformer-based architecture, when GNNs can capture rich structural properties of a layout and require less computation compared to Transformers. This is a very important and interesting experiment, which should be performed.  Moreover, training a Graph Neural Network does not require humungous data. This weakens the motivation of the approach presented in this paper.

---

> ### Author Response · Authors · 2020-11-23
> **Thanks for your comments.**
>
> -Originality.
> We do not present the Transformer details with the assumption that Transformer is a general and popular model for most readers. Thanks for the comments, we would add more descriptions related to our motivation to use such model architecture, and also add more details of the model.
>
>
> -Reference.
> Thanks for the correction. For the first paper you mentioned, it should be “Zheng et al., 2019”, the mistake is made due to our wrong formatting of the latex, we will fix that.
>
>
> -Quality.
>
> 1.2.3. Difference to related works. We will add more descriptions to differentiate our approach from previous works and refine the “Related Work” Section. As you mentioned, one major difference is that we create a large scale of layout dataset and leverage the power of pre-training for general layout representation. We. The average number of layout elements is 6, and the maximum number can be up to 20.
>
> 4. Though the semantic box geometries are considered, we argue that font-size and font-type are also important (for example, large font size might indicate the element is a title instead of a body text).
>
> 5. “Image Captioning” task is to group elements in a canvas which belongs to image caption pair and it requires layout understanding. It has its potential use such as layout recommendation, which group image-text pairs as one element for template mapping.
>
> 6. Since the works you mentioned are related to layout generation, we did not consider them as baselines previously. Yes, as many reviewers have pointed out, we will try to adapt these works as baselines (e.g., use their intermediate outputs).
>
> 7. As for the Graph Neural Network, it would be another promising direction in the next steps, where we need to define relations in the layout as edges.
>
>
> -Clarity.
> Thanks for the suggestions and corrections, we will rephrase some confused descriptions and refine our writing.

---

### Official Review · AnonReviewer4 · 2020-10-29
**A decent work on slide layout representation learning but evaluation can be improved**

**Rating:** 5
**Confidence:** 4

**Review:**

This paper applies state-of-the-art transformer-based neural networks to layout representation learning of slides. The most notable contribution of this paper is the construction of large-scale parsed slide layout dataset. This paper proposes to pre-train the network on this large-scale dataset without masked reconstruction strategy and verifies it with several subtasks including element role labeling, image captioning, auto-completion and layout retrieval, with a comparison to a decision-tree based method as baseline.

+Most of previous layout learning works only show experimental results on small labeled datasets (a few thousands), partially due to the scarcity nature of layout data. This paper looks at slide layout data and constructs a large-scale (>1m) dataset with parsed element properties.
+The chosen network design and training strategies all make sense.

-It is pity that this paper didn’t disclose sufficient details of how the large-scale dataset was constructed and of data statistics, e.g. how many elements in each slide, templates, completeness of properties, etc. How are the properties parsed, fully automatic? Is the role labeling dataset part of the pretraining dataset?
-Pretraining. The proposed evaluation tasks all seem to be sub-tasks of pre-training and it doesn’t look falling into the classic scheme of unsupervised pretraining + supervised fine-tuning. Dataset differentiation is another issue. For example, in the role labeling experiment, is this targeted dataset a subset of the large-scale one? And is the only difference that the training loss?
-Evaluation. Evaluating graphic layout can be a hard problem and this paper tried to propose several small tasks as probes into the learned network. However, it will be more convincing to have a systematic design of experiments. First of all, in addition to type properties, how about geometric property and color property prediction? Second, any experiments would benefit from both quantitative and qualitative results. Especially for layout design, visualization is very important.
-Layout retrieval is an interesting experiment, but manual scoring seems to be arbitrary.
-Baselines. Neural design network by Lee et al. in ECCV2020 and LayoutGAN by Li et al. in ICLR 2019 seem to be good baseline network architectures to compare, although they are trained in different ways.

---

> ### Author Response · Authors · 2020-11-23
> **Thanks for you comments**
>
> -Dataset. We will add more dataset details in the next version. To answer your questions, there are average 6 elements in each slide. Slides are parsed using our internal tool (similar to the publicly available python-pptx library). For the role detection task, the dataset is NOT subset of the pretraining dataset. The role labels are separately defined and annotated.-
>
> -Evaluation. (1) in addition to type, position and color are also useful properties as in some potential downstream tasks, therefore we evaluate them in the layout auto-completion task. We will refine the evaluation to a more systematic approach. (2) Thanks, we will show more visualization case studies in the supplementary. Also we will define more systematic human evaluation. (3) The two papers you mentioned are related to layout generation, but they cannot be directly applied to our tasks related to layout understanding, that’s why we didn’t consider them as baselines. But we will definitely consider to use their intermediate outputs as extra baselines in the next version.

---

### Decision · Program_Chairs · 2021-01-07
**Final Decision**

**Decision:**

Reject

**Comment:**

The paper proposes to learn layout representations for graphic design using transformers with a masking approaching inspired by BERT.  The proposed model is pretrained on a large-collection of ~1M slides (the script for crawling the slides will be open-sourced) and evaluated in several downstream tasks.

Review Summary:
The submission received slightly negative with scores of 4 (R3) and 5 (R2,R4).
Reviewers found the paper to be well-written and clear, and the problem of layout embedding to be interesting.  Reviewers agree that the use of transformers for layout embedding has not been explored in prior work.  However, the paper did not have proper citation and comparisons against prior work for layout embedding, and lacked systematic evaluation.  Reviewers also would like to know more details about the dataset that was used for pre-training.

Pros:
- Novel use of transformers for layout embedding (not yet explored in prior work)
- Use of large dataset of slides

Cons:
- Lacked proper citation and comparisons against prior work for layout embedding
- Lacked systematic evaluation
- Missing details about the dataset

Reviewer Discussion:
During the author response period, the authors responded to the reviews indicating that they will improve the draft based on the feedback, but did not submit a revised draft. As there was no revision to the submission, there was limited discussion with the reviewers keeping with their original scores.  All reviewers agrees that the direction is interesting but the current submission should not be accepted.

Recommendation:
The AC agrees with the reviewers that the current version is not ready for acceptance at ICLR, and it would be exciting to see the improved version.  We hope the authors will continue to improve their work based on the reviewer feedback and that they will submit an improved version to an appropriate venue.